



# Anisotropic transport and frictional properties of simulated clay-rich fault gouges

Elisenda Bakker[1] and Johannes H.P. de Bresser[2]

[1]Department of energy and environment, Royal HaskoningDHV, Laan 1914 35, 3818 EX Amersfoort, The Netherlands
[2]Faculty of Geosciences, Utrecht University, Princetonlaan 4, 3584 CD Utrecht, The Netherlands

*Correspondence to*: Elisenda Bakker (Elisendabakker@hotmail.com)

**Abstract.**

We aimed to evaluate various factors that control the frictional and transport properties of gouge-filled faults cutting carbonate-bearing shales or claystone formations. The research experimentally determined the effect of shear displacement, 10   dynamic shearing, static holding, and effective normal stress on fault gouge permeability, both parallel and perpendicular to the fault boundaries, as well as on frictional behaviour. The simulated gouge was prepared from crushed Opalinus Claystone (OPA), on which we performed direct shear experiments. The direct-shear experiments ($\sigma_n^{eff}$ = 5 - 50 MPa, $P_f$ = 2 MPa, and $T \approx 20°C$) showed ~1 order of magnitude decrease in permeability with shear displacement (up to ~6 mm), for both along- and across-fault fluid flow orientation. Moreover, our data showed an initial, pre-shear permeability anisotropy of up to ~1 15   order of magnitude, which decreased with increasing shear displacement (maturity) to ~0.5, with the along-fault permeability being consistently higher. Our results have important implications for calcite-rich claystones and shale formations, and in particular any pre-existing faults therein, that seal hydrocarbon reservoirs and potential $CO_2$ storage reservoirs, as the current results point to a higher leakage potential of pre-existing faults compared to the intact caprock.

## 1   Introduction

In ambition to lower anthropogenic $CO_2$ emissions, Carbon Capture and Storage (CCS), the capture of $CO_2$ at large point sources followed by injection into geological formations, e.g. hydrocarbon reservoirs or aquifers, is one of the remaining potential methods to mitigate anthropogenic $CO_2$ emissions on a relatively large scale. However, CCS is only a viable option when the long-term sealing integrity of the storage complex, i.e. the reservoir and overlying caprock, can be maintained. Many of the geological storage reservoirs considered for $CO_2$ storage are hydrocarbon reservoirs capped by clay-rich 25   formations, such as the Kimmeridge Clay overlying the Brae Formation of the Miller and Brae fields (Scottish North Sea (Haszeldine et al., 2006)), the Röt or Solling Formations overlying the Triassic Hardegsen (Upper Bunter) Sandstone formations of the P-18 gas field (Dutch North Sea (Samuelson and Spiers, 2012)), and the Tournasian-Viséan mudstones overlying the Carboniferous Tournasian Sandstone unit of the Krechba gas field (Algeria (White et al., 2014)). These clay-rich formations are known to be low permeable rocks (e.g. Armitage et al., 2011; Ingram and Urai, 1999; Neuzil, 1994) and



have proven capacity of structurally trapping hydrocarbons on geological time scales, hence for far longer than required for anthropogenic $CO_2$ storage. However, the potential storage complexes are usually intersected, compartmentalised or bounded by through-going brittle faults. Such faults may represent preferential, naturally occurring, leakage pathways for $CO_2$ during and after injection (e.g. Cappa and Rutqvist, 2011; Hawkes et al., 2005). The integrity of $CO_2$ storage complexes is therefore critically dependent on the sealing capacity and strength of caprock penetrating faults (Knipe et al., 1998).

Brittle fault zones, and specifically clay-rich ones, are structurally anisotropic and lithologically heterogeneous by nature (Caine et al., 1996; Faulkner et al., 2010; Faulkner and Rutter, 1998). These clay-rich fault zones are typically characterised by aligned clay minerals, which are known to weaken the frictional strength of the gouge material and more importantly, in relation to potential $CO_2$ storage, to significantly reduce the gouge permeability to as low as $10^{-22}\,\mathrm{m}^2$, making the fault zone relatively impermeable (Crawford et al., 2008; Takahashi et al., 2007). As such, many faults filled with clay-rich fault gouge

have proven to be efficient seals, acting as structural barriers to across-fault fluid flow (Faulkner and Rutter, 1998; Takahashi, 2003). However, the internal architecture of these fault zones is usually characterised by several distinct structural units, which actually dictate whether the fault zone will indeed act as a structural barrier, or that it behaves as a conduit or combined conduit-barrier system (Caine et al., 1996). The fault architecture usually comprises a fault core surrounded by a pervasively fractured damage zone. Most of the displacement along a fault is accommodated by the fault

core, which is characteristically filled with very fine-grained wear-product, so-called fault gouge. Fault gouge is generally considered to have a low permeability, hence inhibit fluid flow (Takahashi, 2003). Contrary, the surrounding damage zone, may, due to its fractured character, enhance the fault zone permeability, depending on the fracture density and connectivity of the fractures in the damage zone (Mitchell and Faulkner, 2012). In case the fracture density and connectivity of a damage zone is such that a pre-existing fault is likely to form a leakage pathway for fluids, the sealing integrity of the caprock is

considered insufficient and potential reservoirs capped by such seals should not be considered for CCS.

Over time, the sealing capacity of fault gouges within fault cores might be altered by various processes, closely related to the slip behaviour and the stability of a fault gouge. These processes may act on geological time-scales and/or human time-scales. Processes that can alter the fault permeability at geological time-scales include grain size reduction and gouge compaction, as well as dilation. Under static conditions (i.e. unmoving), a fault gouge may exhibit a low permeability,

especially when high normal stresses act on the fault plane (Faulkner and Rutter, 1998). Upon slip, however, a fault gouge may dilate (permeability enhancement) or compact (permeability reduction), depending on the mechanisms controlling deformation of the gouge (Den Hartog and Spiers, 2014). Triggered by human intervention, such as $CO_2$ injection, the same processes that operate on geological time scales may also act on short, human time-scales. The injection of $CO_2$ could potentially lead to 1) a direct increase in pore pressure and associated reduction in effective normal stress acting on pre-

existing fault planes, 2) a pore-elastic response of the storage system, via compaction or heave, or 3) thermal expansion or shrinkage near the well (Vilarrasa et al., 2014; Wang, 2000). As a result of the reduction of normal stress on the fault, the slip reactivation potential and the fault permeability (i.e. leakage potential), which are closely related, will reduce as well (Hooper, 1991; Wiprut and Zoback, 2000; Zoback and Byerlee, 1975). Consequently, the storage system might fail to retain





the injected $CO_2$ in the subsurface. However, only few studies, some in the context of $CO_2$ storage, have attempted to
quantify the direct impact of slip-induced dilation and compaction on the permeability (Im et al., 2018; Samuelson et al.,
2009; Zhang et al., 2007) or studied by means of modelling the coupling of fluid injection-induced fault reactivation (Park et
al., 2020). Furthermore, fault gouge permeability measurements have focussed on measuring across-fault permeability, while
leakage may not only occur from the storage reservoir to adjacent reservoirs, but also along-fault to underlying or overlying
formations (Bense and Person, 2006). Such along-fault leakage may particularly play a role in $CO_2$ storage in aquifers where
an outward pressure gradient may be established following $CO_2$ injection. It should be noted that for $CO_2$ storage, if safe
injection practices are adhered to, such an outward fluid pressure gradient will be unfavourable for promoting along-fault
flow.

In order to predict with some confidence what may happen in faults cutting the caprock of a storage system, it is of great
importance to fully understand what factors control the frictional and transport properties of a fault gouge. In this paper, we
focus on the effect of shear displacement, dynamic shear, static holding and effective normal stress on the permeability, both
parallel and perpendicular to the gouge layer, and the frictional strength of a carbonate-bearing clay-rich fault gouge. To do
so, we report on direct shear experiments in combination with permeability measurements, conducted before, during and
after shear.

## 2   Experimental Method

We performed 11 direct shear experiments in combination with permeability measurements. During these experiments, we
investigate the effect of shear displacement ($x$) and fluid flow orientation on both the transport ($\kappa$) and frictional properties
($\mu$) of a simulated clay-quartz-calcite fault gouge. All 11 experiments have been conducted at room temperature on vacuum-
dried, simulated clay-quartz-calcite fault gouge material at an applied normal stress ($\sigma_n$) of 7 to 52 MPa, using argon gas as
the pore fluid ($P_f$ = 2 MPa). During 4 out of the 11 experiments we performed along-fault permeability measurements before,
during and after shear (Data-set 1). During the remaining 7 experiments we performed across-fault permeability
measurements before, during and after shear (Data-set 2).

### 2.1  Sample material

The clay-rich material used in this study was collected from the "silty-shaly" sub-unit of the Opalinus Claystone formation,
accessible via the Mont Terri Underground Rock Laboratory (URL) (Courtesy of Swisstopo, Dr. C. Nussbaum), and is
described in more detail in Pearson et al. (2003). A permeability measurement conducted on an intact piece of Opalinus
Claystone ($\sigma_n^{eff}$ = 15 MPa) resulted in a permeability value in the order of $10^{-21}$ m$^2$. Simulated fault gouge was prepared by
coarsely crushing small samples of Opalinus Claystone using a pestle and mortar, followed by finer grade crushing using a
ball-mill. The powder was then sieved to obtain the grain size fraction <35μm. X-Ray Diffraction analysis (XRD) performed



on the powdered sample material showed that our material consists of quartz (23%), calcite (26%), pyrite (4%) and
phyllosilicates (47%) (Table 1). We will refer to this gouge as OPA gouge hereafter.

### 2.2 Deformation apparatus

The direct shear experiments described in this study were performed using a conventional triaxial deformation apparatus,
comprising a pressure-compensated main vessel linked to an auxiliary vessel (Figure 1; (Samuelson and Spiers, 2012). The
externally heated, oil-filled main vessel is equipped with a specially-designed direct shear assembly, which can be loaded via
a yoke/piston combination, driven by a motor/gearbox/screwball system, to apply an axial load to the sample assembly,
while simultaneously keeping the system pressure nominally-constant during deformation (resolution ± 0.20 µl). During an
experiment, the confining pressure, the axial load, the displacement of the advancing yoke/piston and sample temperature are
measured using respectively a Jensen pressure transducer (100 MPa range, resolution ± 0.02 MPa), a DRVT-based, semi-
internal load cell (DVRT = Differential Variable Reluctance Transformer; 400 kN range, resolution ± 0.035 kN), a high
precision LVDT (Linear Variable Differential Transformer; 100 mm, ± 0.8 µm) and two K-type (chromel/alumel)
thermocouples (400°C, accuracy ± 0.1 °C, precision ± 0.02 °C). Sliding velocities can be changed step-wise and near
instantaneously, for velocities ranging between ~0.05 and 50 µm/s. The machine is described in detail by Peach and Spiers
(1996) and Hangx et al. (2010a, 2010b). Pore fluids can be introduced into the gouge layer via two stainless steel tubes,
mounted in the load cell block, which are connected to inlets at the top and bottom end pistons inclosing the sample
assembly (Figure 1). We used argon gas (99.9% pure) as pore fluid.
The argon permeability ($\kappa$) of simulated fault gouge was measured using the permeameter previously described by Peach
(1991) and Bakker et al. (2017) (Figure 2). The set-up consists of two symmetrical, low-volume tube/valve systems, located
at both the up and downstream end of the sample assembly, which can be independently pressurized up to 2 MPa, as such
allowing the application of a pressure difference across the gouge layer. The system can be evacuated via a vacuum pump
connected to a vent valve, whereas argon, from a nearby bottle and regulator, is supplied via a second valve. To avoid argon
flow between the sample assembly and the jacket, the applied confining pressure was always maintained higher than the
argon pressure and the gouge layer-piston-jacket contact was taped with Teflon tape. To measure the argon permeability, we
logged the pressure decay across the sample following an imposed pressure difference, using two temperature-compensated
pressure transducers, respectively up- and downstream (2 MPa range, resolution ± 0.001 kPa) (Sutherland and Cave, 1980).
Note, that a very small background leakage of the argon permeameter prevented accurate measurements of samples with
permeabilities of less than $10^{-21}$ m$^2$ (Hangx et al., 2010a).

### 2.3 Sample assembly

For the direct shear experiments described above we used two types of specially designed direct shear assemblies. Both
direct shear assemblies consist of two "L-shaped", direct shear blocks (Figure 3), constituting a 35 mm diameter cylinder
when assembled. Note that given the configuration of the two inverted direct shear blocks, for both sets of direct shear





blocks, the applied normal stress ($\sigma_n$) is equal to the confining pressure ($P_c$) and independent of the shear displacement, at all times during deformation. Each of the two direct shear blocks is equipped with a grooved (60 μm deep, 120 μm wide and 200 μm apart) shear surface (47 mm long and 35 mm wide – Figure 3). To allow for uniform pore fluid access to the gouge layer, via pore fluid channels connected to the tubes in the load cell block, strategically placed porous stainless-steel plates
(permeability >~$10^{-14}$ m$^2$) were placed in the shear surface. The difference between the two sets of direct shear blocks is the location of the porous frits and is described in more detail below.

In preparing each experiment, 3.7 g of loose simulated fault gouge was evenly distributed on the shear interface of the bottom direct shear block and pre-pressed, using a hydraulic press by loading the gouge layer two times for ~30 seconds (~60 seconds in total), at orientations 180° apart (rotation in the horizontal plane). The pre-pressing normal stress was chosen
to be less than 6 MPa, to prevent for overcompaction. The preparation method resulted in coherent, reproducible gouge layers. The bottom block, with the gouge layer (dimensions: 49 mm x 35 mm x ~1 mm – cf. Figure 3) on top, was then covered with the upper shear block, leaving 10-mm gaps at both ends of the gouge layer. The two shear blocks, forming a full cylinder and sandwiching the gouge layer, were then fixated using a heat-shrinking Fluorinated Ethylene Propylene (FEP) inner sleeve, while the 10-mm gaps were filled with 50 μm Teflon foil-wrapped Ecoflex (2-component epoxy) plugs.
To avoid argon flow between the sample assembly and the FEP jacket, rather than through the gouge layer, the FEP-gouge layer-pistons contact was taped off with gas tape. The whole was then placed in a 1.4 mm think Ethylene Propylene Diene Monomer (EPDM) outer jacket, which is sealed against the upper and lower driver blocks using wire tourniquets. The soft Ecoflex plugs, and the FEP and EPDM sleeves are assumed to exert negligible resistance during deformation (Samuelson and Spiers, 2012). Upon jacketing, the sample assembly was attached to the loading frame, emplaced in the main pressure
vessel and evacuated at a Pc of ~5 MPa, for up to ~60 minutes. After evacuation, the appropriate pore fluid was introduced.

Although the direct shear assemblies are similar for the two different types of experiment (along- vs. across-fault), the details of the internal pore fluid systems in the direct shear blocks are slightly different. The direct shear assembly used to measure the across-fault permeability, and frictional strength of the OPA gouge, consist of two stainless steel direct shear blocks (Figure 3a), each equipped with a grooved (60 μm deep, 120 μm wide and 200 μm apart) and porous stainless steel plate
(frit) (effective shear surface: 47 mm long and 35 mm wide and permeability >~$10^{-14}$ m$^2$ – Figure 3d). Uniform pore fluid access to the entire gouge layer is allowed via multiple internal pore fluid channels in the shear blocks (Figure 3a). We will refer to these direct shear blocks as the "across-fault permeability blocks". The along-fault permeability, and frictional strength of the OPA gouge, is measured with the "along-fault permeability blocks", which consists of two titanium shear blocks, each with a grooved (60 μm deep, 120 μm wide and 200 μm apart – Figure 3b) shearing interface. Only 2 mm of the
shearing interface, at the base of each shearing interface, is porous and permeable (permeability ~$10^{-14}$ m$^2$ – frit), allowing pore fluid access to the sample layer from the upper and lower extent of the gouge layer, when assembled. This configuration restricts fluid flow from one end of the gouge layer to the other end through the long axis of the sandwiched gouge layer.



### 2.4 Experimental procedure

For each experiment, evacuation was followed by increasing the $P_c$ to slightly below the pre-determined value, to increase $P_f$

to 2 MPa (using 99.9% pure argon), after which $P_c$ was increased further to the targeted value. After applying the pre-determined $P_c$ and $P_f$-conditions, the system was allowed to equilibrate for ~3h. Subsequently, the downstream part of the permeametry set-up was isolated from the sample and the upstream part (using values "A" and "B" – see Figure 2), to lower the downstream reservoir by 0.2 MPa with respect to the upstream part. After stabilisation, the downstream part was reconnected to the upstream part via valve "B", to allow pore fluid flow from up to downstream, which was measured and

logged as described above. This procedure was repeated for each permeability measurement.

Permeability measurements performed during the experiments were conducted prior to shear, during shear at a sliding velocity of 0.05 μm/s and after a period of hold (referred to as Slide-Hold-Slide (SHS) experiments). The SHS sequence was employed to investigate the effect of static holding versus dynamic shearing on the evolution of fluid flow in a clay-rich gouge layer. The amount of sliding during a permeability measurement depended on the time required to reach full argon

decay. In case full decay was reached within < 2h, (for relatively "high" permeability samples or short path lengths), sliding was continued until full decay was reached, however, if full decay was reached in > 12h ("low" permeability samples or long path lengths), the permeability measurement and sliding were terminated after 4.5h of decay.

A regular slide-hold-slide permeability experiment was typically initiated with measuring the initial permeability ($\kappa_i$ - #1), prior to sliding, followed by a run-in at ~5.4 μm/s. Upon reaching steady state sliding (at ~2 mm), the velocity was reduced

to 0.05 μm/s. Once stable sliding was established (after ~0.02 mm), a dynamic permeability measurement was initiated (#2). Upon full argon decay, the piston was arrested for ~12h, after which the static permeability was measured (#4). Shear was re-induced (at ~5.4 μm/s), reaching steady state shearing after ~ 0.7 mm, before reducing the sliding velocity to 0.05 μm/s. At steady state (after ~0.02 mm), the dynamic permeability measurements was conducted (#4). Subsequently, the piston was halted for ~1h before the static permeability was measured (#5). The experiments were terminated by subsequently halting

and reversing the loading ram at a velocity of 2.0 μm/s until the sample was fully unloaded. Next, the argon pore fluid was removed, followed by removal of the confining pressure, after which the sample assembly was extracted from the apparatus. Following the regular experiments, as described above, the 5 and 25 MPa along-fault samples were sheared for an additional velocity-stepping sequence, sequential to the SHS sequence. Upon the last static permeability measurement, shear was again induced at ~5.4 μm/s (for ~0.7 mm), and reduced to 0.05 μm/s (for ~0.02 mm). This procedure was repeated for a sliding

velocity of 0.1 μm/s, 0.5 μm/s and 1.0 μm/s after which the experiments were terminated and unloaded.

In addition to the regular SHS experiments, supplementary experiments (5 and 50 MPa) were conducted to investigate the effect of hold time on the gouge permeability and frictional strength. These experiments were initiated with an initial permeability measurement (#1), followed by a run-in at ~5.4 μm/s and a reduction in sliding velocity to 0.05 μm/s at ~2 mm. Once a new steady-state was established, a dynamic permeability measurement was initiated (#2). Subsequently, the piston

was halted for 300s and a static permeability measurement (#3) was conducted. This procedure of shearing, measuring and





holding (#4-8), while measuring the permeability was repeated for progressively increasing hold periods of 1000s, 3000s, and ~12h prior to terminating the experiment. In case >1 mm of shear displacement was left, a last cycle of shearing and holding, for a hold time of ~1h, was applied before termination.

### 2.5 Data acquisition and processing

During each experiment, the internal axial load, piston displacement, confining pressure, sample temperature and argon pore pressure decay were logged every 2 s using a 16-bit National Instrument A/D converter and Labview VI Logger system. To obtain the shear stress ($\tau$) and shear displacement ($x$), the raw data was processed. Elastic machine distortion effects were corrected for using pre-determined, polynomial stiffness calibrations performed at or near the pressure and temperature conditions applied in this study to obtain true axial displacement values. For the used assembly configuration, the shear

stress is equal to the internal axial load divided by the contact area of the shear surface, which is assumed to remain equal to the initial contact area during the experiment.

#### 2.5.1 Permeability calculations

The argon permeability $\kappa$ was calculated using the transient-pressure decay method described by Sutherland and Cave (1980). To calculate $\kappa$, a pressure difference $\Delta P_0$ [MPa] is applied between two gas reservoirs ($V_1$, $V_2$ [m$^3$]), located up- and

downstream of the gouge layer, and allowed to equilibrate with time $t$ [s] through the layer over length $l$ [m] and across-sectional area $A$ [m$^2$], following the exponential pressure decay equation (Sutherland and Cave, 1980):

$$\Delta P = (\Delta P_0)e^{-\alpha t} \tag{1}$$

With

$$\alpha = \kappa \left(\frac{A}{l}\right)\left(\frac{V_1+V_2}{V_1V_2}\right)\left(\frac{1}{\mu\beta}\right) \tag{2}$$

Yielding

$$\kappa = \frac{d}{dt}\left(\ln(\frac{\Delta P}{\Delta P_0})\right)\frac{l}{A}\left(\frac{V_1V_2}{V_1+V_2}\right)\delta\beta \tag{3}$$

where $\delta$ represents the dynamic viscosity of argon gas at room temperature [Pa s] and $\beta$ the argon compressibility [Pa$^{-1}$]. The permeability is then determined from the linear slope fitting $\ln(\frac{\Delta P}{\Delta P_0})$ vs. $t$ using an in house developed computer program, READ30 (Peach, 1991). The permeability is flow path length dependent. As such the path length for the "across-fault" permeability ($\kappa_{\perp}$) is equal to the thickness of the gouge layer, whereas for the "along-fault" ($\kappa_{//}$) permeability the path length

is the long axis of the gouge layer. For the across-fault permeability calculations, we have to assume that the thickness of the gouge layer is equal to the final thickness at all stages in the experiment, as the current set-up does not allow us to measure the change in layer thickness during the experiment. For the along-fault permeability calculations, we assumed the changes in path length to be equal to the shear displacement at the various stages in the experiment.





The permeability is a measure of the mobility of a fluid within a medium, and as such is the property of the medium and not
of the fluid (Klinkenberg, 1941). Under steady-state and laminar flow conditions ("Darcy flow"), the measured (apparent)
permeability of a medium is therefore considered to be (close to) the true (intrinsic) permeability of a medium, for any kind
of fluid (Klinkenberg, 1941). In Darcy flow, molecule-molecule collisions dominate fluid interactions, therefore molecule-
wall collisions are neglected. However, when the pore radius of a medium decreases, approaching the mean free path of the

fluid molecules, the frequency of molecule-wall collisions increases, resulting in gas slippage along the pore walls. This
effect is known as the "slip effect" or "Klinkenberg effect". The true permeability can be derived from the apparent
permeability when corrected for this slippage effect by using:

$$K_a = K(1 + \frac{b}{\bar{p}}) \hspace{4cm} (4)$$

where $K_a$ [m$^2$] is the apparent permeability, $K$ [m$^2$] the intrinsic permeability, $b$ is the Klinkenberg constant and $\bar{p}$ the mean

pressure. As a consequence, extrapolation of the apparent permeability to infinite pressure ($\frac{1}{p} = 0$) then gives the true
permeability.

## 3   Results

All direct shear experiments, experimental conditions and key data, including $\mu_{ss}$ at ~2.0 mm shear displacement, are listed in
Table 2. We adopt the convention that compressive stresses are positive and take the effective normal stress $\sigma_n^{eff}$ as the

applied confining pressure, hence normal stress $\sigma_n$, minus the applied pore pressure $P_f$, yielding:

$$\sigma_n^{eff} = \sigma_n - P_f \hspace{4cm} (5)$$

### 3.1 Permeametry data

The evolution of the along- and across-fault permeability is plotted, together with the friction coefficient, as a function of
shear displacement in Figures 4b-i. The initial, pre-shear permeability, measured at effective normal stresses $\sigma_n^{eff}$ of 5 to 50

MPa and $T \approx 20°C$, ranges between $1.2*10^{-15}$ (at 5 MPa) and $1.6*10^{-17}$ m$^2$ (at 50 MPa) for the along-fault orientation (Figure
4b-f) and $2.3*10^{-16}$ (at 5 MPa) and $1.8*10^{-18}$ m$^2$ (at 50 MPa) for the across-fault orientation (Figure 4g-i). Note that the initial
permeability measured along-fault is 2 to 7 times higher than measured for the across-fault orientation (Table 2). All
experiments show a rapid change (i.e. decrease) in permeability in the first ~2 mm of shear displacement, down to $2.8*10^{-16}$
(5 MPa – 77% decrease) and $8.4*10^{-19}$ m$^2$ (50 MPa – 95% decrease) for the along-fault orientation and $1.5*10^{-16}$ (5 MPa –

33% decrease) and $3.4*10^{-19}$ m$^2$ (50 MPa – 85% decrease) for the across-fault orientation. With increasing shear
displacement, i.e. shear strain, all experiments show a continuous, though less substantial change in permeability, compared
to that observed in the first 2 mm of shear displacement, reaching near-constant values towards the end of the experiments
(Figure 4). In this range, the along-fault permeability is only 1 to 3 times higher than the across-fault permeability (Table 2).
Overall, the permeability values obtained for the along-fault orientation are consistently higher than obtained for the across-





fault orientation, though the difference seems to diminish with increasing shear displacement (see permeability ratios in Table 2).

Detailed analysis of the obtained permeability measurements shows, superimposed on the downward change in permeability with shear displacement, a reduction in permeability for each subsequent period of hold or sliding (Figure 4). Our data furthermore shows that the difference in permeability is <20% with respect to the preceding measurement, irrespectively if

that it is a measurement after hold or during shear. One exception to this trend is observed in the 17.5 MPa-across-fault experiment, where the third permeability measurement, during shear (Figure 4d), showed a relative increase in permeability with respect to the preceding value obtained during hold.

Besides a decrease in permeability with shear displacement, our results show a decrease in permeability with increasing effective normal stress, irrespectively of orientation. This is explicitly shown in Figure 5a, where the permeability $\kappa$ is

plotted as a function of effective normal stress. Moreover, Figure 5a shows that the decrease in permeability is larger for an increase in effective normal stress from 5 to 50 MPa than for ~2-6 mm of shear displacement.

The initial, pre-shear permeability is plotted as a function of the inverse mean pore pressure in Figure 5b, for measurements obtained at a $\sigma_n^{eff}$ of 5, 10 and 17.5 MPa in order to assess the Klinkenberg effect. At effective normal stresses of 5 and 10 MPa, the difference between the obtained apparent (gas) permeability for a mean pore fluid pressure of 2 MPa (cf. $1/P_m =$

0.05 MPa$^{-1}$) and the determined true (intrinsic) permeability is <20%. At an effective normal stress of 17.5 MPa, the apparent (gas) permeability, for a mean pore fluid pressure of 2 MPa, deviates ~90% from the true permeability (Figure 5b).

Measurements of the initial, pre-shear permeability (#1) of the same sample always show a reproducibility within 4%, in most cases even to within 1%. The reproducibility of the evolution of permeability with shear displacement (#2-#8) between different experiments performed under the same conditions ($\sigma_n^{eff} = 5$ or 50 MPa, $P_f = 2$ MPa and $T \approx 20°C$) is less good, as

illustrated in Figure 6. To rule out that the observed increase in permeability with increasing shear displacement in 22OPAPA5 (Figure 6a), is the result of gas leakage along the outer boundaries of the gouge layer, we continued to perform experiments with tape closing off that potential leakage pathways (see sample assembly).

### 3.2 Frictional behaviour

The friction coefficient ($\mu$) versus displacement curves obtained for the simulated, argon-saturated OPA gouges are included

in Figures 4b-i. All experiments show a rapid increase in $\mu$ until apparent yielding (~0.43-0.73), occurring at a displacement of ~0.3-1.2 mm and occasionally followed by limited slip-hardening reaching a peak strength and a drop in $\mu$, before reaching (near) steady-state friction coefficients of 0.54 to 0.91 at ~2 mm of shear displacement (Table 2). Moreover, they show a modest decrease in friction coefficient following the imposed reduction in sliding velocity.

The steady state friction coefficient obtained after re-shear is generally lower than, or equal to, the pre-hold steady state

value. Furthermore, the majority of the friction curves is characterised by (near) strain-neutral behaviour following the run-in stage, with the exception of the 25 and 50 MPa across-fault curves, which show minor strain-hardening (Figure 4).





The frictional strength decreases, independently of orientation, with increasing effective normal stress, from max. ~0.9 to as low as 0.5. However, this decrease is non-linear, which is explicitly shown in Figure 7a for the steady state frictional strength at ~2 mm as a function of effective normal stress. Generally, stable sliding was observed for all sliding velocities, however during the 0.05 μm/s sliding intervals of the 5 MPa and 25 MPa across-fault experiments, plus the 5 MPa along-fault experiment (i.e. 27OPAPA5t, 30OPAPC25t and 31OPAPC5t), stick-slips were observed (see in-set in Figure 4h). Similarly, the additional velocity-steps following the regular slide-hold-slide sequence of 5 and 25 MPa across-fault experiments (i.e. 30OPAPC25t and 31OPAPC5t) showed stick-stick behaviour for the 0.05 to 0.5 μm/s interval. By plotting the steady state frictional strength obtained at ~5.4 μm/s and 0.05 μm/s, complemented by the friction coefficient values obtained for the additional sliding velocities, the frictional strength is plotted as function of sliding velocity in Figure 7b for the 5 and 25 MPa across-fault samples (i.e. 31OPAPA5t and 30OPAPC25t). Although stick-slip behaviour was observed in the friction curves, an increase in frictional strength with increasing sliding velocity is observed in Figure 7b.

The effect of the different hold time sequences employed in this study on the frictional and transport properties of OPA gouge used in the slide-hold-slide experiments is illustrated in Figure 6 for two sets of duplicate experiments employing increasing hold time (along-fault – at $\sigma_n^{eff}$ = 5  MPa and across-fault at $\sigma_n^{eff}$ = 50  MPa). Moreover, one experiment employed a 12h hold, following the initial sequence (across-fault – at $\sigma_n^{eff}$ = 5 MPa – Figure 6b). Comparing Figures 6a with 6b, and 6c with 6d, no systematic effect emerges of increasing the hold time from 300 to 1000 and finally 3000 seconds on the friction coefficient and permeability. The results in case of a different hold time history, from 12 to 1 hour hold (Figure 4g versus Figures 6a and 6b), neither show reason to infer such an effect.

The shear stress ($\tau$) supported at steady-state at ~2.0 and at 3.4- and 4.5-mm shear displacement, is plotted as a function of the $\sigma_n^{eff}$ in Figure 7 for all (argon-saturated) experiments performed. The data show a linear relation passing through or close to the origin, suggesting nearly cohesionless fault gouges (Figures 7c-d). Fitting the Mohr-Coulomb criterion, i.e. $\tau = \mu\sigma_n^{eff} + C_0$, to the linear $\tau$ vs. $\sigma_n^{eff}$ data yields very similar cohesions ($C_0$) of roughly 2 MPa for both along- and across-fault orientation (Table 3), when considering the error bars. The cohesion furthermore appears to be independent of shear displacement, as the cohesion remains to be ~2 MPa (Figure 7c-d – Table 3).

## 4    Discussion

One of the major concerns of $CO_2$ injection in geological formations is the potential risk of inducing fault slip and (micro)seismicity along previously sealing (i.e. impermeable) gouge-filled faults (Hawkes et al., 2005; Zoback and Gorelick, 2012). Fault slip and induced seismicity are most likely the result of a change in the stress conditions on pre-existing faults. This can occur via 1) (local) compaction or heave of the reservoir (poro-elastic effect), 2) thermal expansion or shrinkage near the well, or 3) an increased pore pressure in a pre-existing fault, effectively reducing the normal stress on the fault





(Hawkes et al., 2005; Wang, 2000). Once fault slip is induced, this may cause dilatation, potentially increasing fault
permeability. Upon slipping, leakage of $CO_2$ could then either occur across the fault, i.e. between two adjacent reservoir
compartments, or along the fault, e.g. into an over- or underlying formation, in case an outward pressure gradient is in place
(Bense and Person, 2006).

To help evaluate the potential for leakage along or across gouge-filled faults cutting clay-quartz-calcite caprock, we have
investigated the effect of shear displacement, dynamic shear, static holding, and normal stress on the transport properties
(both along- and across-fault) of a simulated clay-quartz-calcite fault gouge.

In the following, we will first discuss our findings regarding the effects of shear displacement, dynamic shear, static holding,
and effective normal stress on the permeability evolution, along- and across-fault, of a simulated clay-quartz-calcite fault
gouge. We will then discuss the implications for sealing integrity and frictional behaviour of a gouge-filled fault cutting a
potential clay-quartz-calcite caprock in $CO_2$ storage systems.

### 4.1 The effect of shear displacement, dynamic shear, static hold, and normal stress on fault gouge permeability: across- and along-fault

In discussing our experimental data on gouge permeability, we start with the notion that, as shown in Sect. 3.1, hold periods
of different time spans and different sequences, do not show a systematic effect on the permeability or friction coefficient.
This justifies the grouping of experiments with different hold histories, including both along- and across-fault permeability
measurements, in one figure as done in Figure 4. We note that Im et al. (2018) found a systematic change in permeability
with hold time for Westerly granite and for Green River shale (containing more than 50% carbonate). In particular the
absolute permeability enhancement for the granite was substantial, while it was small for the shale. The effect on shale
becomes better visible if using a normalized permeability increase. Nevertheless, the results of Im et al. (2018) show that
differences in material properties play a role, which likely explains the difference with our observation regarding hold time
and permeability change.

Overall, our data show a decrease in permeability with shear displacement (i.e. fault maturity) and with effective normal
stress (Figure 5a) for both along- and across-fault orientation. The most significant reduction in permeability is observed in
the first few millimetres of shear displacement, reaching (near) constant values during further shear displacement. Over a
shear displacement of up to ~6 mm, the permeability changed by up to an order of magnitude compared with the pre-shear
permeability. Comparing the along- and across-fault permeability measurements demonstrates an initial anisotropy of up to 1
order of magnitude. Furthermore, our permeability results, both along- and across-fault, show that the decrease in
permeability with increasing displacement coincides with a decrease in frictional strength (Figure 4). Note also that the
friction coefficient decreases with increasing effective normal stress (Figure 7a).

The observed sharp decrease in permeability, characteristic for the initial stage of shear, followed by a more gradual decrease
towards more or less constant values with continuous shear is reported for a broad range of clastically-derived compositions
(e.g. Morrow et al., 1984; Zhang and Tullis, 1998; Bakker et al., 2017). Furthermore, the total decrease of about one order of




magnitude upon shear, both along- and across-fault, is in agreement with permeability measurements on mixtures rich in phyllosilicates, including chlorite, illite, kaolinite, montmorillonite and muscovite (Figure 4 – e.g. Crawford et al., 2008; Morrow et al., 1984; Zhang et al., 1999). Besides the reduction in permeability with shear displacement, we observe a
reduction in permeability with increasing effective normal stress, again in agreement with many previous studies (e.g. Behnsen and Faulkner, 2012; Crawford et al., 2008; Morrow et al., 1981b; Zhang et al., 1999).

During initial loading, prior to shear, mineral alignment of the clay-quartz-calcite mixtures is assumed to be dominated by rotation of phyllosilicates towards an orientation perpendicular to the highest principal stress orientation, i.e. the applied normal stress, hence parallel to the boundaries of the gouge layer (Zhang et al., 2001). As a consequence, fluid flow across
the fault is more tortuous than along the fault, which is supported by the permeability anisotropy observed for the pre-shear permeability values (Figure 5a). The sharp, initial decrease in permeability during the run-in stage of the experiments is inferred to reflect shear-enhanced compaction of the bulk gouge layer, as rearrangement of grains and grain size reduction allow for a denser grain packing (e.g. Marone and Scholz, 1989; Zhang et al., 2001). Upon reaching steady-state shearing, thin and dense shear bands are expected to have developed accommodating localised shear (Haines et al., 2009), which will
not affect the bulk permeability of the gouge layer any further during subsequent shear displacement, dynamic shear and/or periods of holding. This is supported by the transition to more or less constant permeability values for the bulk of the gouge layer obtained during this stage of the experiment. Moreover, the permeability anisotropy shows a decrease with respect to the initial permeability anisotropy. This suggests a shear-induced reduction in tortuosity contrast between along- and across-fault fluid flow, in agreement with observation for mica-gouges by Zhang et al. (1999), who inferred the reduction to result
from an anisotropic enhancement of fluid flow by P-shear and Y-shears. Permeability measurements obtained during dynamic shear and static holding show little difference, suggesting a limited effect of holding versus shearing on the permeability, especially when a well-defined foliation is established (Figure 4).

The difference in permeability and permeability evolution observed for the various effective normal stresses (Figure 5a), i.e. high permeability values at low effective normal stress and lower values at higher normal stresses (Figure 4), suggests
various degrees of compaction via grain rearrangement and grain crushing (Zhang and Tullis, 1998). At the same time, the observed decrease in frictional strength with increasing effective normal stress suggests an indirect correlation between the friction coefficient and permeability. A similar relation has been observed between the mineral composition and permeability by Crawford et al. (2008). However, we have too little experimental data to convincingly confirm such correlation with friction coefficient. What also plays a role is the fact that at high effective normal stress (>17.5 MPa), the Klinkenberg effect
cannot be ignored. Given the specific experimental procedure employed in this study, a systematic Klinkenberg correction (Eq. (4)) could not be applied, as during shear the maturity of the fault, and therefore the experimental situation, continuously changed. As a consequences, observed trends of increasing or decreasing permeability are meaningful, but not too much importance can be attributed to absolute values, at least not at effective normal stress above 17.5 MPa. More work is needed on these aspects.





### 4.2 The effects of deformation conditions on the frictional behaviour of simulated clay-quartz-calcite fault gouges


In discussing our friction results, obtained for the (argon-saturated) OPA samples (see Sect. 3.1), we begin by noting that the effect of calculated cohesion of ≤2MPa (Figure 7; Table 3) is relatively high at low effective normal stresses (<25 MPa), and relatively small at higher effective normal stress (>25 MPa). Therefore, we infer that the shear strength of our samples,

expressed as friction coefficient, can be represented by the ratio of the shear stress over the effective normal stress at effective normal stresses ≥25 MPa, especially at $\sigma_n^{eff}$ = 50 MPa as these values are most relevant for CCS in geological reservoirs. This inference is further supported by the fairly good agreement between the experimentally derived friction coefficients at 25 and 50 MPa and the internal friction coefficient at yield or peak, derived via Eq. (3) (Table 3). However, the cases that the agreement is less good convey the samples performed at effective normal stresses < 25 MPa. These should

therefore be assessed carefully.

Our friction data showed higher friction coefficient values at lower effective normal stresses than at 50 MPa (Figure 4). A similar gradually decreasing friction coefficient to more constant values at higher effective normal stresses has been observed for various phyllosilicate-bearing mixtures (Behnsen and Faulkner, 2012; Ikari et al., 2007; Saffer and Marone, 2003), and is assumed to reflect the influence of gouge cohesion at low effective normal stresses as mentioned above.

In addition to the observed change in absolute frictional strength with changing effective normal stress, we also observed an increase in friction coefficient with increasing velocity, and vice versa, for the mineral compositions and deformation conditions tested (Figures 4 and 6). Despite the positive dependence of friction coefficient on velocity seen in the experiments (Figure 4), it is striking that low amplitude, stick-slip behaviour was observed in several of the (argon-saturated) OPA samples. Specifically, stick-slip was observed in samples 27OPAPA5t, 31OPAPC5t and 30OPAPC25t, which were

sheared at room temperature, at sliding velocities of 0.05-1.086 μm/s and effective normal stresses of respectively 5, 5, and 25 MPa. Based on the Rate and state slip theory stick-slips are not expected in gouges exhibiting velocity-strengthening behaviour. It is unlikely that this behaviour is a chemical effect of the argon gas, given its inert nature. There is also no systematic correlation between the observed stick-slip behaviour and across- versus along-fault argon flow. Possible explanations that we can propose for the "anomalous" stick-slip include the following:

1) Local, stepwise penetration of argon into previously unpenetrated shear bands, due to shear band dilatation at the low effective normal stresses used in the experiments showing stick-slip. This could be possible without affecting sample scale permeability if it is transport through the body of the sample that determines its permeability.

    2) We cannot exclude the possibility mentioned earlier that small amounts of water may be introduced into the sample during argon injection, from traces left in the pore fluid system after earlier wet experiments. Gradual penetration of

410        water introduced into the gouge in this way, could conceivably lead to slip-weakening and associated stick-slip. Some support for this is offered by the correlation between slip-weakening and stick-slip seen in the relevant argon-saturated experiments.



3) A further possibility is that the samples showing stick-slip at $\sigma_n^{eff} = 5$ MPa became over-consolidated, with respect to shear-testing, due to pre-pressing during preparation. This would produce a peak strength during shear followed

by slip-weakening (Ide and Takeo, 1997) and hence potential for unstable slip events.

## 5  Implications

In order to predict with some confidence what may happen in faults cutting the caprock of a potential $CO_2$ storage system, it is of importance to understand what factors control the frictional and transport properties of a gouge-filled fault. The goal of this research was therefore to evaluate the effect of shear displacement, dynamic shear, static holding and effective normal

stress on i) fault gouge permeability, both parallel and perpendicular to the fault boundaries, and ii) on the frictional strength and stability of a simulated clay-rich gouge. Both aims are particularly relevant for, but not limited to, $CO_2$ injection and storage in potential storage reservoirs in the subsurface as for both aims different aspects of fault integrity are evaluated. The implications of our results are listed below:

1) Faults intersecting a clay-quartz-calcite-bearing formation, such as a calcite-rich claystone or shale, may be (re-

)activated upon hydrocarbons production or $CO_2$-storage, potentially leading to induced (micro-)seismicity, fault zone dilation, permeability enhancement and as a consequence a reduction of sealing capacity. It is therefore critical to know how shear displacement, dynamic shear, static holding and effective normal stress affect the along- and across-fault permeability. Assuming a clay-quartz-calcite caprock composition resembling the Opalinus Claystone (40-50% phyllosilicates, 20-25% quartz, and 20-30% of calcite) the present results imply that with increasing shear

displacement the permeability will reduce up to an order of magnitude, with the most significant reduction occurring in the first millimetres of shear displacement. With increasing maturity, upon reaching steady-state frictional behaviour, faults will acquire a well-developed internal foliation, which has only limited potential for compaction or dilation. As a consequence, re-shear or static holding will only slightly decrease the permeability further, before levelling off to near constant permeability.

2) Potential $CO_2$ storage reservoirs are located at 1-4 km depth (cf. 25-100 MPa). The present results (Figure 4) show that pre-existing gouge-filled faults at these depths are expected to have permeabilities ranging between $10^{-16}$ and $10^{-19}$ m$^2$ (Klinkenberg uncorrected). Comparing these values with the permeability of an intact slice of Opalinus Claystone ($7.7 \cdot 10^{-20}$ m$^2$ (Klinkenberg uncorrected) at 15 MPa effective normal stress or ~0.7 km depth) shows that, even at normal stresses corresponding to 2 km, the gouge permeability of $10^{-19}$ m$^2$ is still 1-2 orders of magnitude

higher than the intact rock, and therefore more likely to act as a leakage pathway.

3) If the pore fluid pressure in a fault zone increases due to migration of supercritical $CO_2$ from a storage reservoir, the effective normal stress can decrease. Our results show that with increasing pore pressure, and decreasing effective normal stress, the permeability of simulated clay-quartz-calcite fault gouge will increase, irrespectively of fluid flow orientation, reducing sealing capacity. Moreover, our results show that leakage along a fault into an over- or





445          underlying formation is easier than leakage across the fault into neighbouring reservoir compartments, with our

simulated fault gouges showing ~1 order of magnitude anisotropy in permeability.

### 6    Conclusions

In this study, we aimed to evaluate various factors that control the transport and frictional properties of gouge-filled faults cutting carbonate-bearing shale or claystone formations. The research experimentally determined the effect of shear

displacement, dynamic shearing, static holding, and effective normal stress on i) fault gouge permeability, both parallel and perpendicular to the fault boundaries, and ii) on the frictional strength and stability of a simulated clay-rich gouge. To achieve this, direct shear experiments in combination with along- and across-fault permeability measurements were performed. The direct shear experiments were performed at effective normal stresses of 5 to 50 MPa on simulated fault gouges prepared from Opalinus Claystone (OPA). Our key findings are given below:

1)  The Klinkenberg uncorrected, initial fault permeabilities for the (unleached) Opalinus Clay gouges, measured at room temperature, fell in the range of $10^{-15}$ (5 MPa) to $10^{-17}$ m$^2$ (50 MPa) along the fault, versus $10^{-16}$ (5 MPa) to $10^{-18}$ m$^2$ (50 MPa) across the fault. Both along- and across-fault permeabilities (Klinkenberg uncorrected) decreased with shear displacement, showing the largest drop in permeability in the first ~2 mm of shear displacement. At this stage, a well-developed internal foliation has developed,

with the permeability decreasing to near constant values. Permeability decreased further with shear, and during static hold periods, via compaction of the fault gouge material. Up to an order of magnitude decrease in permeability is observed for up to 6 mm shear displacement, in both fluid flow orientations. Pre-shear permeability values show up to an order of magnitude difference (anisotropy) between along- and across-fault, with shear displacement this difference decreases.

2)  Pre-existing gouge-filled faults with a clay-quartz-calcite composition located at a depth range of 1-4 km are expected to have permeabilities ranging between $10^{-16}$ and $10^{-19}$ m$^2$ (Klinkenberg uncorrected). The Klinkenberg uncorrected permeability of the intact rock ($7.7 \cdot 10^{-20}$ m$^2$ at 15 MPa effective normal stress or ~0.7 km depth) is, even at normal stresses corresponding to 2 km, 1-2 orders of magnitude lower than the fault gouge. Gouge-filled faults are therefore more likely to act as a leakage pathway.

3)  The permeability of simulated clay-quartz-calcite fault gouge will increase, thereby reducing the sealing capacity, irrespectively of fluid flow orientation, if the pore fluid pressure in a fault zone increases due to migration of supercritical $CO_2$ from a storage reservoir. Moreover, leakage along a fault into an over- or underlying formation is easier than leakage across the fault into neighbouring reservoir compartments (~1 order of magnitude anisotropy in permeability).



4)   Frictional strength decreases with increasing effective stress, illustrating an effect of cohesion on the resistance to shear observed at low effective normal stress (<25 MPa).

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



**Table 1. Mineralogical composition of simulated Opalinus Claystone fault gouges (OPA) given in percentages (%). The uncertainty is given as the mean value of the difference in percentages of multiple batches analysed.**

| Mineral | Proportion in OPA (%) |
|---|---|
| Quartz | 21 |
| Calcite | 17 ± 3 |
| Pyrite | 2 |
| Muscovite | 33 ± 11 |
| Illite | 23 |
| Kaolinite | 12 ± 1 |
| Chamosite/chlorite | 7 |


**Table 2. List of experiments performed and experimental conditions. Name: OPA = Opalinus Claystone, P = permeability, A = along-fault, C = aCross-fault, t = tape along the outside of the sample assembly. Symbols: T = temperature, σneff = effective normal stress, Pf = pore fluid pressure, V = sliding velocity, Th = holding time, μss = steady-state friction coefficient at ~2.0 mm, μ~3.4 = friction coefficient at ~3.4 mm, μ~4.5 = friction coefficient at ~4.5 mm.**

| Experiment | T (•C) | $\sigma_n^{eff}$ (MPa) | V (µm/s) | $T_h$ (s/h) | $\mu_{ss}$ (~2.0 mm) | $\mu_{\sim 3.4}$ | $\mu_{\sim 4.5}$ | Permeability ratio (along vs. across) |
|---|---|---|---|---|---|---|---|---|
| *Permeability measurements at T = room temperature and $P_f$ = 2 MPa* | | | | | | | | |
| 15OPAPC50 | 22 | 50 | 5.43 – 0.0543 (3 x) | 300s, 1000s, 3000s | 0.54 | 0.52 | - | 7.1-2.5-1.2 |
| 25OPAPA25t | 27 | 25 | 5.43 – 0.0543 (2 x) | ~12 h | 0.58 | 0.55 | 0.52 | |
| 26OPAPA10t | 22 | 10 | 5.43 – 0.0543 (2 x) | ~12 h, ~1h | 0.72 | 0.68 | 0.65 | |
| 27OPAPA5t | 23 | 5 | 5.43 – 0.0543 (2 x) | ~12 h, ~1h | 0.91 | 0.77 | 0.88 | 5.2-1.8-1.8-1.9 |
| 28OPAPA17.5t | 23 | 17.5 | 5.43 – 0.0543 (2 x) | ~12 h | 0.63 | 0.60 | 0.58 | |
| 29POPAPA50t | 23 | 50 | 5.43 – 0.0543 (1 x) | ~12 h | 0.50 | 0.50 | 0.49 | |
| 30OPAPC25t | 22 | 25 | 5.43 – 0.0543 (2 x) 0.0543 – 0.1086 – 0.543 – 1.086 | ~12 h, ~1h | 0.48 | 0.46 | 0.47 | 2.2-1.5-1.6-1.6 |
| 31OPAPC5t | 22 | 5 | 5.43 – 0.0543 (2 x) 0.0543 – 0.1086 – 0.543 – 1.086 | ~12 h, ~1h | 0.70 | 0.68 | 0.69 | |
| *Additional direct shear experiments as shown in Figure 6* | | | | | | | | |
| 7OPAPC50 | 22 | 50 | 5.43 – 0.0543 (3 x) | 300s, 1000s, 3000s | 0. | 0. | - | |
| 22OPAPA5 | 23 | 5 | 5.43 – 0.0543 (3 x) | 300s, 1000s, 3000s | 0.77 | 0.77 | - | |
| 23OPAPA5t | 22 | 50 | 5.43 – 0.0543 (4 x) | 300s, 1000s, 3000s, ~12 h | 0.80 | 0.86 | - | |



**Table 3. List of internal friction coefficient and cohesion values obtained for the permeability experiments.**

|  | Internal friction coefficient (-) | Cohesion (MPa) |
|---|---|---|
| Along fault at ~2.0 mm | 0.47 ± 0.011 | 2.1 ± 0.31 |
| Along fault at ~3.4 mm | 0.48 ± 0.011 | 1.7 ± 0.30 |
| Along fault at ~4.5 mm | 0.47 ± 0.017 | 1.8 ± 0.55 |
| Cross fault at ~2.0 mm | 0.48 ± 0.042 | 0.76 ± 1.7 |
| Cross fault at ~3.4 mm | 0.47 ± 0.033 | 0.63 ± 1.34 |

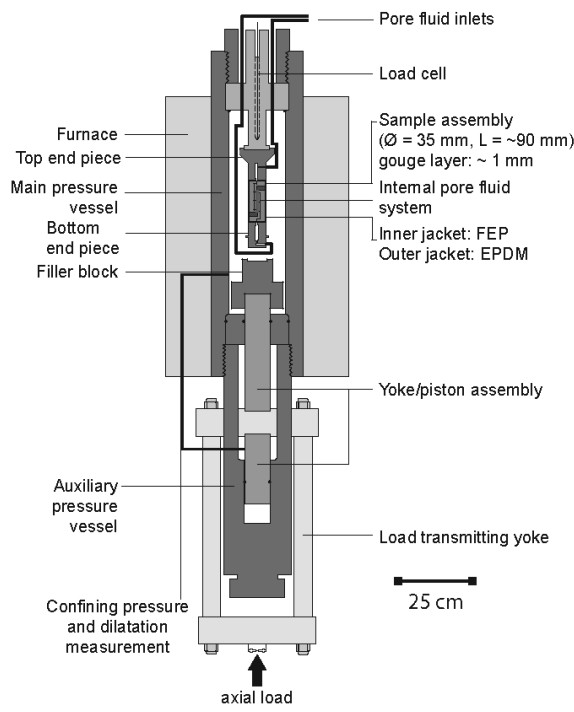


**Figure 1. Schematic diagram of the triaxial deformation apparatus used in the present study (modified after Hangx et al., 2010).**



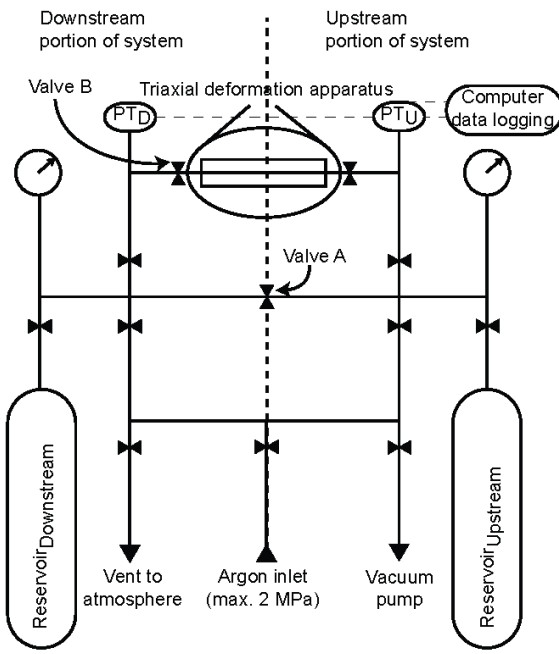

**Figure 2. Schematic representation of the argon permeameter set-up used in this study in conjunction with the triaxial deformation apparatus.**





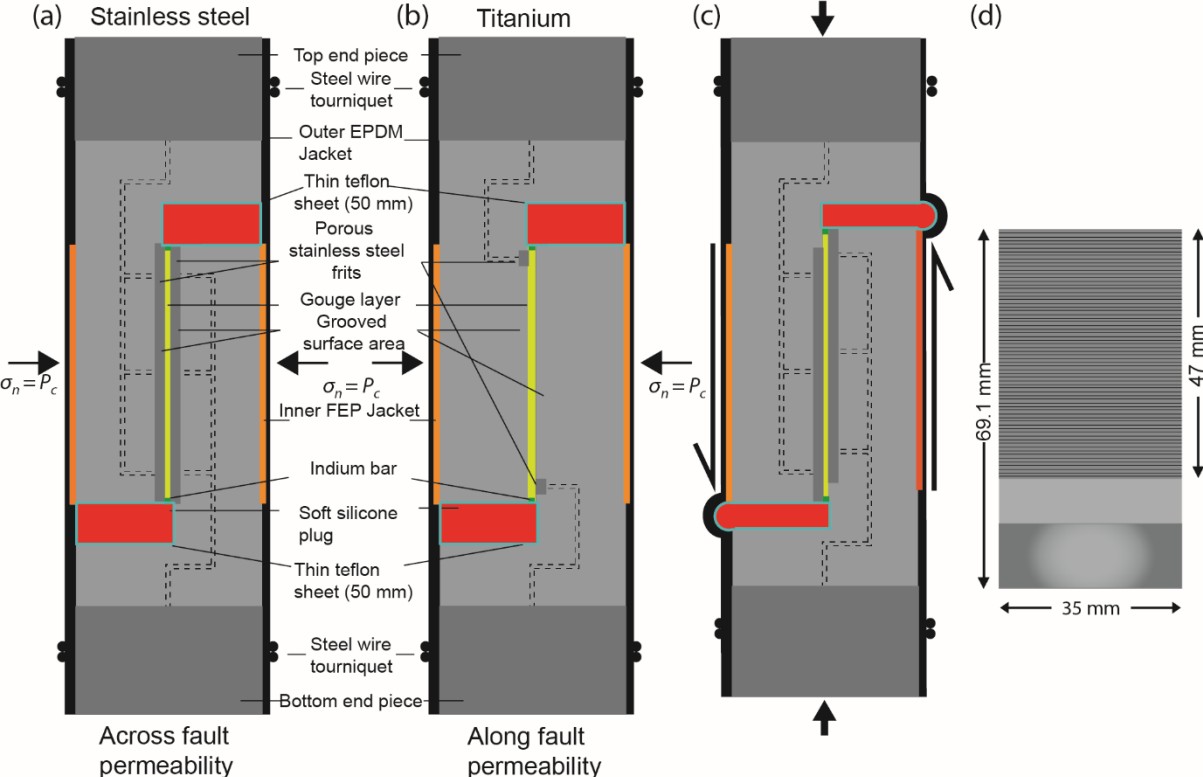


**Figure 3. Schematic representation of the inverted shear assemblies DSB-1 (across-fault) and DSB-2 (along-fault). a and b) Pre-shear schematic of inverted shear blocks (grey), fluid distribution frits (dark grey), internal pore fluid system (dashed lines), gouge layer (yellow), indium bars (green), FEP inner jacket (orange), EPDM outer jacket (black), silicone putty plugs (red) and thin teflon sheet (light blue). The blocks are assembled with a relative offset, creating gaps filled with the silicone plugs, to provide**
**space for the necessary displacement distance. Between the two pistons a ~ 1mm thick simulated gouge layer is sandwiched. c) Post-shear configuration of the inverted DBS-1 shear blocks. Note that during the experiment, the silicone putty (wrapped in teflon sheets - light blue) becomes compressed and extrudes. d) Schematic drawing of the active face a shear block, showing the toothed porous frit which grips the sample (modified after Samuelson and Spiers, 2012).**






**Figure 4.** Permeability evolution (orange dots connected by grey tie-line) and frictional strength (solid blue line) as a function of shear displacement observed for the slide-hold-slide direct shear experiments (Data-set 1 & 2) conducted at room temperature, σneff = 5-50 MPa and Pf = 2 MPa. a) Schematic illustration of fluid flow orientation. b-f) Permeability and frictional strength evolution as a function of shear displacement for the along-fault experiments. g-i) Permeability and frictional strength evolution as a function of shear displacement for the across-fault experiments.






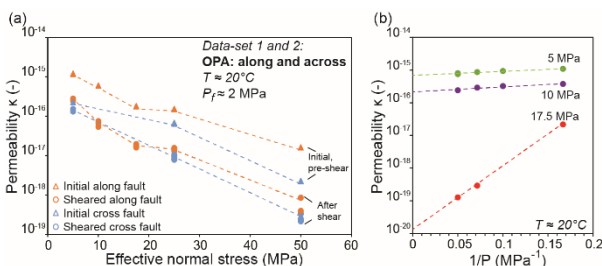

**Figure 5. a) Permeability as a function of effective normal stress for the initial, pre-shear permeability, and during and after shear permeability. b) Evolution of permeability as a function of the inverse mean pore pressure at an effective normal stress of 5, 10 and 17.5 MPa.**

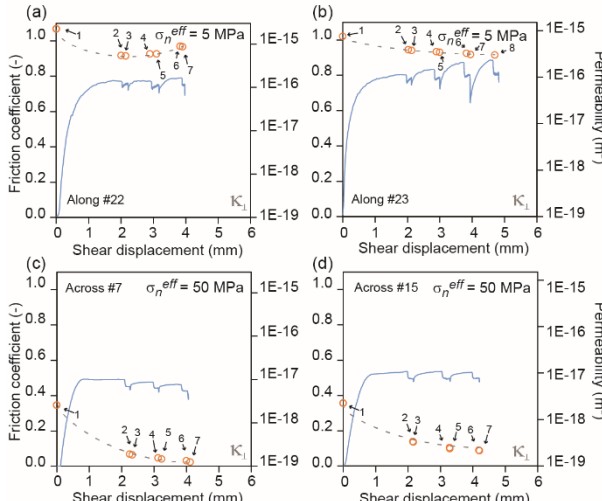

**Figure 6. Additional direct shear experiments with along-fault permeability (a and b; respectively 22OPAPA5 and 23OPAPA5t) and across-fault permeability measurements (c and d, respectively 7OPAPC50 and 15OPAPC50) and a slide-hold-slide sequence with increasing hold periods of 300, 1000, 3000 seconds (a, c, d) and additionally 12 h (b) to investigate the effect of hold duration on the decrease in permeability, tested at room temperature. a and b) Permeability and frictional strength evolution as a function of shear displacement for tested at $\sigma_{neff}$ = 5 MPa, $P_f$ = 2 MPa. c and d) Permeability and frictional strength evolution as a function of shear displacement tested at $\sigma_{neff}$ = 50 MPa, $P_f$ = 2 MPa.**

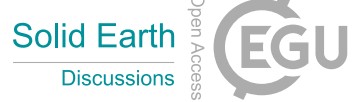

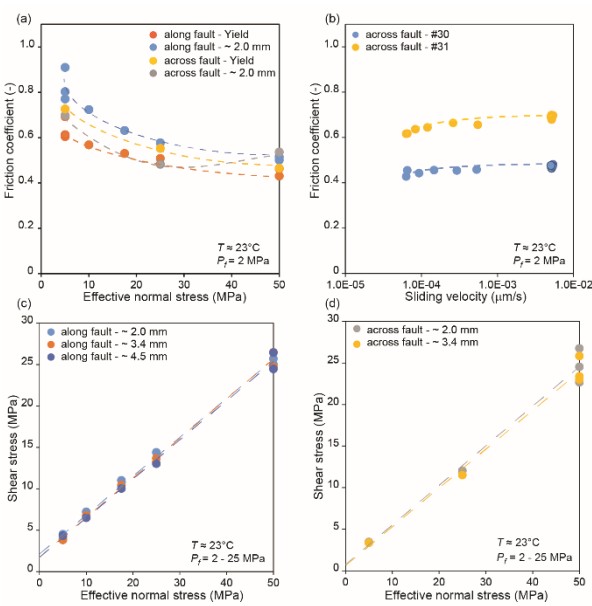

**Figure 7. Shear stress at steady-state, i.e. at ~2 mm, 3.4 and 4.5 mm shear displacement, as a function of effective normal stress.**
**The dashed linear trend lines and corresponding data points are colour-coded. a) For experiments in which the along-fault**
**permeability is measured. b) For experiments in which the across-fault permeability is measured. c) Friction coefficient at yield**
**and at steady-state (~2 mm) as a function of effective normal stress. d) Friction coefficient at steady-state for the various sliding**
**velocities used for the along- and across-fault orientation.**
