# Peer review of "Anisotropic transport and frictional properties of simulated clayrich fault gouges"

_Solid Earth, 2020_

## Referee Comment (RC1) · Anonymous Referee #1 · 22 Jan 2021

The paper by Bakkers and de Bresser presents new experimental data on the influence of pressure and slip on permeability (and permeability anisotropy) of clay-rich gouge. The new data are certainly valuable, and those experiments are not easy to conduct. The paper is framed heavily around CO2 storage problems, which is just one of the many possible applications of the work; this is somewhat distracting, since a lot of space is devoted (in the introduction and discussion) on the link with CO2 storage, which can only be addressed very generically here, at the expense of more physical/microstructural discussions about the processes that generate changes in permeability.

[Figure]

My main concern with the paper is twofold. Firstly, it is not clear at all if the data are useable, since the author repeatedly mention that they could not achieve a proper correction for the Klinkenberg effect. This should be clarified. Secondly, the data interpretation in terms of mechanisms (in absence of other measurements, such as pore volume, or microstructures, or modelling attempts) are very vague and rely almost entirely on comparison and analogies with published literature.

In that sense, the paper remains very technical and descriptive. One key question that is not really addressed, for instance, is the role of pre-compaction: if permeability is anisotropic to start with, it means that normal compaction of the layer already produces a texture, which is then altered by shear. But how is this initial texture formed? Is it visible in the microstructure? Is this an experimental artefact or something that we should expect in nature?

One thing that could help put the results in perspective with literature data is the make systematic comparison between the permeability anisotropy data obtained here and the other existing datasets. Is there anything general that can be established? What is the order of magnitude of anisotropy that we should expect across the range of materials tested?

Detailed comments:

l.29: permeable -> permeability

l.56: not sure that dilation and compaction (i.e., porosity change) can be so easily linked to increase of decrease in permeability. Maybe moderate the statement?

l.66: one recent reference that is relevant here is Rutter and Mecklenburgh, JGR 2018, where systematic characterisation of shear and normal stress effects on permeability anisotropy was conducted. Also, Okazaki, Katayma and Noda, GRL, 2013, specifically studied along-fault permeability vs slip in a phyllosilicate gouge.

l.81: by "transport", it seems that the authors mean "permeability". (Transport is more

vague and could refer to hydraulic diffusivity)

l.175: does "dynamic permeability" refer to "syn-deformation permeability"? the word "dynamic" means different things to different people, and some clarity would bring everyone on the same page.

l.177: maybe reword as "... a dynamic permeability measurement was conducted"

l.209: the symbol on the lhs seems to be "proportional to" instead of "alpha".

l.217: is there a way to estimate the potential error produced by this assumption? I don't think it will be huge, but a rough estimate could be helpful, since compaction might lead to artificial increase in permeability (reduction in A).

l.243: just say "... a rapid decrease"?

l.255: remove first "it"

l.270: I do not really understand what was done here, and what was the conclusion. Should we trust the data or not?

l.350: is the reduction in k with increasing sigma_n reversible? how much of that is elastic closure of pathways vs. permanent collapse?

l.370: I understand that the measurement of volumetric strain (or pore volume change) was not possible in the experiments. Would there be a way to back the statements in the discussion using some indirect observations (say, microstructures, or anything else), rather than just relying on literature data (on other rocks!)?

l.375: What exactly is the problem here? I am not I follow what is stated.

l.389: I do not understand the sentence; rephrase?

l.397-415: I am not convinced that such small anomalies deserve to be discussed at such great length, especially since no real explanation is given beyond speculation. Is the stick-slip behaviour reproducible under those conditions?

l.430: this conclusion strongly depends on the state of consolidation of the gouge layer. How do the laboratory condition reflect the in-situ conditions of natural faults? In nature, faults might be overconsolidated or chemically sealed, which would lead to dilation (not compaction) upon slip.

l.455: I am not sure what it means to show "uncorrected" values. They may not be meaningful at all, unless some reasonable error bars can be provided. Are the results upper or lower bounds for the actual permeability? In addition, if $CO_2$ is the focus, the interesting permeability value is that relevant to (possibly gaseous) $CO_2$, which then implies another Klinkenberg effect.

l.459: foliation is mentioned but not shown? here again, microstructures would be important to support this point.

l.470: I sense that this point, as stated, could be made independently from the data shown in the paper. I am not convinced this is a solid conclusion that is drawn from the new dataset, rather than a generic point about permeability.

---

## Referee Comment (RC2) · Anonymous Referee #2 · 26 Feb 2021

In this paper, the authors describe a series of direct shear experiments conducted on simulated gouge materials made of crushed Opalinius Claystone. They aim at investigating the mechanical behavior and gas conductivity evolution of caprock penetrating faults. The paper is well organized and written in a clear manner. However, the authors fail to convey the originality of their work and the relevance to real application in my opinion. Moreover, some of the interpretation of the results are not very convincing.

1. Other authors in the literature have already looked at the behavior of clay-rich fault gouge varying the amount of clay content, the normal pressure, etc. and performed permeability tests during the experiments. As stated in the present paper, the results

obtained in these previous studies (e.g. Crawford et al. 2008 or Zhang et al., 1999) are similar to the ones obtained here. Therefore, it is not clear the new information brought by this study.

2. The mechanical behavior of clays is strongly dependent on the presence of water, and it is likely that a fault zone will be saturated in natural conditions. Therefore, the influence of water on the results obtained here and how it should modify what would happen in the field should be addressed by the authors.

3. The authors used an interesting setup to induce flows across or along the experiments. However, the way the samples are prepared should not lead to any anisotropy for the permeability (compressed powder of fine particles). Therefore, the anisotropy they have observed (especially for the initial permeability) appears more as an artifact of the experiment, coming from flow path across the gouge that is more subjected to heterogeneities compared to the flow along the gouge. It makes the comparison with the anisotropy of fault zones not very justified.

4. A mature fault zone presenting a gouge is supposed to have accommodated a large amount of displacements and reach the "critical state" (as defined in the soil mechanics community). At this stage, the material does not change its volume when sheared, so the volume variation observed here and affecting the permeability in the first millimeters of the experiment would presumably not be observed.

5. The fact that the fault zone as a conduit along its shearing direction is mostly due to the damage zone that you are not studying.

6. The interpretation of the evolution of friction with normal stress should be elaborated. The decrease of the friction with increasing effective normal stress is attributed to be an effect of the cohesion, whereas this effect has been extensively studied in Rock mechanics and is related to the change in volumetric behavior. Several models like Hoek-Brown or Cam-clay have been specifically developed to describe this effect. Stick slip coming from slip weakening. . .

7. The observation of stick-slips events during the experiments presented here are quite interesting as the material is velocity-strengthening, but the possible explanation to explain this behavior are not convincing. Argument 1 seem quite difficult to verify and does not seem very likely. Arguments 2 and 3 (lines 405-415) mention some mechanisms that would induce some slip-weakening. If there was any slip weakening it should be observed on the mechanical response of the material. Moreover, slip-weakening leads to a single instability (or stress drop) and cannot create repetitive stick-slip events.

---

## Author Comment (AC1) · 21 Apr 2021

P. de Bresser Anonymous Referee #1

The paper by Bakkers and de Bresser presents new experimental data on the influence of pressure and slip on permeability (and permeability anisotropy) of clay-rich gouge. The new data are certainly valuable, and those experiments are not easy to conduct. We thank the reviewer for pointing out the value of the new data and appreciate the
recognition of the complexity of our experiments.

The paper is framed heavily around $CO_2$ storage problems, which is just one of the many possible applications of the work; this is somewhat distracting, since a lot of space is devoted (in the introduction and discussion) on the link with $CO_2$ storage, which can only be addressed very generically here, at the expense of more physical/microstructural discussions about the processes that generate changes in permeability. We indeed focus on faults penetrating caprocks of $CO_2$ storage systems, realizing of course that the outcome has wider applications than only $CO_2$ systems. However, the nature of our experiments, namely shear tests with measurement of argon gas permeability, makes that the results help determining (see also the response to referee 2) the upper limit of the fault gouge permeability/frictional strength and trends with increasing shear displacement and increasing normal stress. In other words, it is the limits and trends rather than the absolute values that form the main output. And these may help assessment of leakage risks. So we prefer to keep the framing to $CO_2$ storage systems.

My main concern with the paper is twofold. Firstly, it is not clear at all if the data are useable, since the author repeatedly mention that they could not achieve a proper correction for the Klinkenberg effect. This should be clarified. This point should be viewed in the same light as point 2 of referee 2. With the study we aimed to investigate the trend in permeability with increasing shear displacement and increasing normal stress, for carbonate rich clay materials relevant for caprocks of reservoirs. Similar as to Faulkner and Rutter (2000), and many more studies, we observe a decreasing trend with increasing shear displacement, and normal stress, as well as an anisotropy between perpendicular and parallel to fault. Ideally, we would have preferred to establish a proper Klinkenberg correction, to account for differences in permeability of the gouge to gas as that to water. However, as explained in the paper, a systematic correction could not be applied due to the progressive change of the maturity of the fault during shear. A Klinkenberg correction would have affected the absolute values for permabil-

ity, lowering the values relative to those observed in or study, but the current results still give the upper limit of the fault gouge permeability/frictional strength and support the trends with increasing shear displacement and increasing normal stress also found by others (on different type of materials).

Secondly, the data interpretation in terms of mechanisms (in absence of other measurements, such as pore volume, or microstructures, or modelling attempts) are very vague and rely almost entirely on comparison and analogies with published literature. In that sense, the paper remains very technical and descriptive. One key question that is not really addressed, for instance, is the role of pre-compaction: if permeability is anisotropic to start with, it means that normal compaction of the layer already produces a texture, which is then altered by shear. But how is this initial texture formed? Is it visible in the microstructure? Is this an experimental artefact or something that we should expect in nature? We recognize the observation of the reviewer that the paper is rather technical and relies on a comparison with literature rather than using own microstructural observations. Unfortunately, it appeared not possible to retain deformed samples such that meaningful microstructural analysis was possible. Regarding the role of pre-compaction, see our response to comment 3 of referee 2. The permeability anisotropy develops due to the physical nature of the play clay minerals, that rotate to an orientation with their basal planes tending to become parallel to shear/fault plane.

One thing that could help put the results in perspective with literature data is the make systematic comparison between the permeability anisotropy data obtained here and the other existing datasets. Is there anything general that can be established? What is the order of magnitude of anisotropy that we should expect across the range of materials tested? This is a good suggestion, and we will include such comparison in the revised manuscript. We will focus on studies that have looked into permeability anisotropy, and compare the effect of effective normal stress, composition, shear displacement, dry or wet permeability measurements and the effect of shear and maybe holding on the permeability development. These studies include [Zhang et al., 1999;

Crawford et al., 2008; Okazaki et al., 2013]. Materials used in these studies vary from pure kaolinite/muscovite/serpentinite to pure quartz, and quartzo-feldspatic to granitic rock, allowing us to make general inferences to which we can compare the outcomes of our experiments. One important aspect is that the order of magnitude of anisotropy depends on the nature of the grains. Based on work by Zhang et al. (1999) we know that in quartzo-feldspathic gouges, the initial anisotropy is small, however after some sliding a significant anisotropy develops. The quartz gouges have shown to have no significant anisotropy to begin with and that increases with shear, up to a maximum of one order of magnitude. With permeability perpendicular to shear being the lower. The same is observed for granitic material. Resulting in a permeability anisotropy of 1.5 orders. Okazaki et al., 2013 observed a similar trend, resulting in a permeability anisotropy of ∼1. Muscovite on the other hand [Zhang et al., 1999], starts with an initial anisotropy of about 1 order of magnitude. Shearing leads to a decrease in permeability for both directions, however the decrease in along fault permeability is lower, resulting in a smaller anisotropy. This observations is similar to what we have observed with our mixed composition gouges. Text will be modified to include the above in a systematic way.

Detailed comments: l.29: permeable -> permeability Done

l.56: not sure that dilation and compaction (i.e., porosity change) can be so easily linked to increase of decrease in permeability. Maybe moderate the statement? We can do this

l.66: one recent reference that is relevant here is Rutter and Mecklenburgh, JGR 2018, where systematic characterisation of shear and normal stress effects on permeability anisotropy was conducted. Also, Okazaki, Katayma and Noda, GRL, 2013, specifically studied along-fault permeability vs slip in a phyllosilicate gouge. The paper by Rutter and Mecklenburg, 2018, did indeed look at the effect of normal and shear stress on the hydraulic transmissivity of shale, among other rock materials. However, the paper is focussed on thin cracks within/between two rock parts, rather than at fault gouges

like we do in our study. However, we will give credit to this study. We will also add the suggestion of Okazaki et al., 2013. They have studied the shear-induced permeability anisotropy of a simulated serpentinite gouge and saw a shear-induced permeability anisotropy of ∼1 order.

l.81: by "transport", it seems that the authors mean "permeability". (Transport is more vague and could refer to hydraulic diffusivity) Done

l.175: does "dynamic permeability" refer to "syn-deformation permeability"? the word "dynamic" means different things to different people, and some clarity would bring everyone on the same page. Changed

l.177: maybe reword as "... a dynamic permeability measurement was conducted" Done

l.209: the symbol on the lhs seems to be "proportional to" instead of "alpha". Done

l.217: is there a way to estimate the potential error produced by this assumption? I don't think it will be huge, but a rough estimate could be helpful, since compaction might lead to artificial increase in permeability (reduction in A). The reduction in A plays a role in the calculation of the permeability along fault. As stated in the paper, we assumed the changes in path length to be equal to the shear displacement at the various stages in the experiment for the along-fault permeability calculations. As such, we have based our reduction in A on the shear displacement. In this way thickness changes do not affect changes in A.

l.243: just say "... a rapid decrease"? Unclear what this is referring to.

l.255: remove first "it" Done

l.270: I do not really understand what was done here, and what was the conclusion. Should we trust the data or not? To avoid argon leakage along the pistons, so to not present a "short-circuit" route for fluid flow outside the gouge layer, we taped the set-up on the outside. This contained the argon gas to the gouge layer, resulting in accurate

permeability measurements.

l.350: is the reduction in k with increasing sigma_n reversible? how much of that is elastic closure of pathways vs. permanent collapse? Based on work done by e.g. Faulker and Rutter, 1998, who tested the effect of pressure cycling, the reduction in permeability with increasing confining pressure/ effective pressure is dominated by permanent collapse. With each consecutive pressure cycle they observed a reduction in permeability. This was attributed to the reshuffling of the phyllosilicates with respect to one another, producing enhanced compaction with each pressure cycle. However they also observed that there is a slight difference in permeability with complete or partial depressurization of the sample, for which they suggested the opening or re-opening of microcracks (during complete depressurization). But they assumed that this effect is negligible at higher pressures.

l.370: I understand that the measurement of volumetric strain (or pore volume change) was not possible in the experiments. Would there be a way to back the statements in the discussion using some indirect observations (say, microstructures, or anything else), rather than just relying on literature data (on other rocks!)? No, unfortunately we are not able to back it with microstructures, as these were damaged due to their fragile nature when preserved with epoxy.

l.375: What exactly is the problem here? I am not I follow what is stated. It is not clear to us where this is referring to?

l.389: I do not understand the sentence; rephrase? We will do this.

l.397-415: I am not convinced that such small anomalies deserve to be discussed at such great length, especially since no real explanation is given beyond speculation. Is the stick-slip behaviour reproducible under those conditions? See our response to comment 7 of referee 2. We will follow the suggestion to downsize the discussion given the fact that the anomalies are relatively small..

l.430: this conclusion strongly depends on the state of consolidation of the gouge layer. How do the laboratory condition reflect the in-situ conditions of natural faults? In nature, faults might be overconsolidated or chemically sealed, which would lead to dilation (not compaction) upon slip. Unfortunately, we don't really understand this statement. The conclusion as derived in our paper is based on the fact that our experiments show a reduction in permeability with increasing shear displacement, which is the equivalent of increasing fault maturity, and with static holding and re-shearing. These situations happen in nature are well, i.e. relatively long hold periods leading to consolidation interrupted by short and rapid periods of shearing, all while being under natural stresses. Yes, natural faults might be overconsolidated and might be chemically sealed, but neither are required for a natural fault to be a fault. As for overconsolidation requires an abnormal stress regime and chemical alteration requires oversaturated fluids running through a fault.

l.455: I am not sure what it means to show "uncorrected" values. They may not be meaningful at all, unless some reasonable error bars can be provided. Are the results upper or lower bounds for the actual permeability? In addition, if $CO_2$ is the focus, the interesting permeability value is that relevant to (possibly gaseous) $CO_2$, which then implies another Klinkenberg effect. The values documented in this paper represent the upper bounds for clay-rich, quartz-calcite fault gouge permeability. Argon is chosen as medium as it is an inert gas, as such it will not cause any interaction with the fault gouge. Any possible interaction of gas with the fault gouge material, such as clay swelling, will lead to a further reduction of the permeability. As such the argon permeability values represent the upper bound values. As these values point to near impermeable values, $CO_2$ or even water in the fault zone are expected to lower the permeability values even more. Making leakage even less probable.

l.459: foliation is mentioned but not shown? here again, microstructures would be important to support this point. As we agree with the referee that microstructures would be important to support this point, we were not able to preserve the microstructures in

epoxy due to the fragile nature of the gouge.

l.470: I sense that this point, as stated, could be made independently from the data shown in the paper. I am not convinced this is a solid conclusion that is drawn from the new dataset, rather than a generic point about permeability. This statement is based on our experiments which show that the permeability will be higher for lower effective normal stresses. As this is inversely proportional with an increasing pore fluid pressure as a result of the migration of supercritical $CO_2$ from a storage reservoir into a fault. Moreover, our data show that flow along a fault into over- or underlying formations is easier than into neighbouring reservoir compartments. We could rephrase this point such as to make it clearer that this is point is derived from the data obtained in this study.

---

## Author Comment (AC2) · 21 Apr 2021

P. de Bresser Anonymous Referee #2

In this paper, the authors describe a series of direct shear experiments conducted on simulated gouge materials made of crushed Opalinius Claystone. They aim at investigating the mechanical behavior and gas conductivity evolution of caprock penetrating faults. The paper is well organized and written in a clear manner. We appreciate this

comment on the organization and clarity of the paper.

However, the authors fail to convey the originality of their work and the relevance to real application in my opinion. Moreover, some of the interpretation of the results are not very convincing.

1. Other authors in the literature have already looked at the behavior of clay-rich fault gouge varying the amount of clay content, the normal pressure, etc. and performed permeability tests during the experiments. As stated in the present paper, the results obtained in these previous studies (e.g. Crawford et al. 2008 or Zhang et al., 1999) are similar to the ones obtained here. Therefore, it is not clear the new information brought by this study. The originality of the paper lies in the use of a fault gouge during deformation experiments prepared from a natural rock that is directly relevant when studying faults in caprocks of reservoirs. Many caprocks of (former) hydrocarbon reservoirs that are now considered for CO2 storage are clay-rich, hence the choice of the natural Opalinus Claystone as our source material. Previous studies used mainly artificially composed mixtures: Crawford et al. (2008), for example, made synthetic mixtures of fine-grained quartz and kaolinite, Zhang et al. (1999) used pure quartz, feldspar and muscovite gouges. The latter authors also made granitic gouge from natural Westerly granite, but that is not a rock type widely relevant as reservoir caprock. Moreover, our Opalinus claystone not only is a mixture of quartz and clay, but also contains a substantial amount (∼17%) carbonates. Previously tested mixtures only contained trace amounts of carbonate (e.g., Rutter and Mecklenburg, 2017). Finally, we investigated the characteristics of the fault gouge during and after shear under conditions relevant for CCS. Our results thus are complementary to results obtained in previous studies on fault gouges studied because of their role in the behaviour of major lithospheric faults such as the San Andreas fault (e.g. SAFOD project).

2. The mechanical behavior of clays is strongly dependent on the presence of water, and it is likely that a fault zone will be saturated in natural conditions. Therefore, the influence of water on the results obtained here and how it should modify what would

happen in the field should be addressed by the authors. Undoubtedly, water is important in natural fault zones. But our study has to be seen in the context of CCS and the possibility that a fault becomes flooded with CO2 gas, as would be the case in a leakage scenario, resulting in drying out the fault gouge. Two aspects are important. First, previous permeability studies [e.g. Faulkner and Rutter, 2000] carried out with both water (wet) and argon (dry) have shown that water permeability values are consistently lower than argon gas permeability values. Secondly, wet fault gouges have been found to show lower friction coefficients than dry gouges (e.g. Morrow et al, 2000; Moore and Lockner, 2004). Shear tests with measurement of argon gas permeability, as carried out in our study, thus help determining the upper limit of the fault gouge permeability/frictional strength and trends with increasing shear displacement and increasing normal stress (which is related to/representing an increasing depth). That being said, we can add to the manuscript that the influence of water on the results presented by us is that the permeability during increasing shear displacement is likely to become very low, close to impermeability and thus to cause a fault to act as a barrier to flow.

3. The authors used an interesting setup to induce flows across or along the experiments. However, the way the samples are prepared should not lead to any anisotropy for the permeability (compressed powder of fine particles). Therefore, the anisotropy they have observed (especially for the initial permeability) appears more as an artefact of the experiment, coming from flow path across the gouge that is more subjected to heterogeneities compared to the flow along the gouge. It makes the comparison with the anisotropy of fault zones not very justified. The natural rock used to prepare the gouges from is a claystone with a high percentage, close to 47%, of platy minerals (phyllosilicates). During the preparation of the fault gouge layer, the pre-pressing perpendicular to the (future) shear plane, at normal stress below 6 MPa, will already cause the platy minerals to rotate to an orientation perpendicular to the maximum stress, hence towards parallelism with the shear/fault plane. This will be further enhanced during testing at normal stresses of 10 MPa and higher. As a result, an initial permeability anisotropy develops prior to shear. Such anisotropy has also been observed in other studies, such as that of Zhang et al. (1999) on pure muscovite gouges. These authors found an order of magnitude difference in permeability parallel and perpendicular to the experimental fault plane before actual shearing. This was attributed to progressive alignment of the basal planes of muscovite to the plane of the fault. Although our samples are not 100% clay, our close to 50% volume of platy minerals is high enough to result in a measurable anisotropy. The development of a planar fabric is well known from natural fault gouges containing clay (e.g. Faulkner and Rutter, 2000 and references therein). Hence, the permeability anisotropy in our samples is not an unwelcome artefact of our experiments; on the contrary, it makes comparison of experiments with nature more meaningful.

4. A mature fault zone presenting a gouge is supposed to have accommodated a large amount of displacements and reach the "critical state" (as defined in the soil mechanics community). At this stage, the material does not change its volume when sheared, so the volume variation observed here and affecting the permeability in the first millimetres of the experiment would presumably not be observed. The largest decrease in permeability indeed takes place roughly the first 2 mm of shearing. The gradual change in permeability continues during further shearing beyond 2 mm, though less substantial. We have not been able to measure volume changes so cannot relate permeability changes to volume changes during specific parts of the progressive shearing. It is known though from other studies (e.g. Faulkner and Rutter, 2000) that permeability changes occur without further compaction or with only very minor dilatation. We thus consider the changes in permeability in our experiments, and in particular the systematic difference between along fault and across fault, as related to the progressive development of an internal foliation with increasing shear displacement. This is directly relevant to the broad aim of this paper, obtaining improved insight into the evolution of faults in clay-rich caprocks.

5. The fact that the fault zone as a conduit along its shearing direction is mostly due to

the damage zone that you are not studying. It is true that our study does not include the damage zone, which of course is due to the character of the experimental set-up, with a gouge layer between two stainless steel pistons. This is not different from several previous studies with comparable set-ups. So our results are of importance for the fault core, and then it depends on the architecture of the fault zone as a whole to what extent differences between fault core and damage zone control overall permeability. Cain et al (1996) present a conceptual model of the permeability structure in fault zones, with end members defined by the relative percentage of core (localized conduit vs. localized barrier) and of the relative percentage of the damage zone (distributed conduit vs. combined conduit-barrier). In case you have a low damage zone with a low to high fault core width, a fault can act as a localised conduit or as a localised barrier. Our results are relevant for these architectures.

6. The interpretation of the evolution of friction with normal stress should be elaborated. The decrease of the friction with increasing effective normal stress is attributed to be an effect of the cohesion, whereas this effect has been extensively studied in Rock mechanics and is related to the change in volumetric behavior. Several models like Hoek-Brown or Cam-clay have been specifically developed to describe this effect. Stick slip coming from slip weakening. To our knowledge, the effect of cohesion the reviewer is referring to is more widely studied and know from experiments under low normal stress (kPa range to a maximum of $\sim$4 MPa), so in soil mechanics, than it is known from (friction) experiments on rock materials at normal stresses in the MPa range. At steady state sliding at such normal stresses, in principle no volume change will be expected, and the effect of cohesion (absolute value low relative to the normal stress) will be limited. We have looked into the Hoek-Brown-PAC and Cam-Clay models. The Hoek-Brown-PAC model provides a representation for yielding that accounts for the changing failure condition. The modified Cam-Clay model is an incremental hardening/softening elastoplastic model. We will refer to these models in the revised manuscript, with special attention to the implied volumetric effects. However, we are limited by the fact that our experimental set up did not allow monitoring of volume changes.

7. The observation of stick-slips events during the experiments presented here are quite interesting as the material is velocity-strengthening, but the possible explanation to explain this behavior are not convincing. Argument 1 seem quite difficult to verify and does not seem very likely. Arguments 2 and 3 (lines 405-415) mention some mechanisms that would induce some slip-weakening. If there was any slip weakening it should be observed on the mechanical response of the material. Moreover, slip weakening leads to a single instability (or stress drop) and cannot create repetitive stick-slip events. The stick slip events that we observed are relatively small anomalies, as also indicated by reviewer 1 (see reviewer's comment l.397-415). We did not observe the events in all samples, and have not systematically investigated them. For that reason, we can only speculate what the explanation is for the events. We decided to include a short description of the events since we do not want to ignore the observations, though not really the focus of our work. We will follow the suggestion of referee 1 who indicates the anomalies should not be discussed at great depth.